# Polar pattern formation induced by contact following locomotion in a multicellular system

**Masayuki Hayakawa[1], Tetsuya Hiraiwa[2,3], Yuko Wada[1], Hidekazu Kuwayama[4], Tatsuo Shibata[1]\***

[1]Laboratory for Physical Biology, RIKEN Center for Biosystems Dynamics Research, Kobe, Japan; [2]Mechanobiology Institute, National University of Singapore, Singapore, Singapore; [3]Universal Biology Institute, University of Tokyo, Tokyo, Japan; [4]Faculty of Life and Environmental Sciences, University of Tsukuba, Tennodai, Ibaraki, Japan

**Abstract** Biophysical mechanisms underlying collective cell migration of eukaryotic cells have been studied extensively in recent years. One mechanism that induces cells to correlate their motions is contact inhibition of locomotion, by which cells migrating away from the contact site. Here, we report that tail-following behavior at the contact site, termed contact following locomotion (CFL), can induce a non-trivial collective behavior in migrating cells. We show the emergence of a traveling band showing polar order in a mutant *Dictyostelium* cell that lacks chemotactic activity. We find that CFL is the cell–cell interaction underlying this phenomenon, enabling a theoretical description of how this traveling band forms. We further show that the polar order phase consists of subpopulations that exhibit characteristic transversal motions with respect to the direction of band propagation. These findings describe a novel mechanism of collective cell migration involving cell–cell interactions capable of inducing traveling band with polar order.

**\*For correspondence:**
tatsuo.shibata@riken.jp

**Competing interests:** The authors declare that no competing interests exist.

## Introduction

The collective migration of eukaryotic cells plays crucial roles in processes such as wound healing, tumor progression, and morphogenesis, and has been the focus of extensive study (*Haeger et al., 2015*). The collective effects are typically associated with cell–cell interactions, such as long-range interaction mediated by secreted chemicals or short-range stable cohesive interaction mediated by adhesion molecules. However, the study of self-propelled particles in physics has revealed that motile elements which lack such activities may nonetheless give rise to dynamic collective motion, such as a traveling band (*Chaté et al., 2008*; *Ginelli et al., 2010*; *Ohta and Yamanaka, 2014*; *Solon et al., 2015*), mediated by a relatively simple transient short-range interaction, such as alignment interaction (*Marchetti et al., 2013*; *Vicsek et al., 1995*; *Vicsek and Zafeiris, 2012*). The emergence of such collective motions of self-propelled particles, such as formations of clusters and traveling bands, has been observed in a wide variety of systems, ranging from animal flocks (*Ballerini et al., 2008*), bacteria swarms (*Wioland et al., 2013*; *Zhang et al., 2010*), and cell assemblies (*Szabó et al., 2006*) to biopolymers and molecular motors (*Butt et al., 2010*; *Schaller et al., 2010*; *Sumino et al., 2012*). For cell assemblies of eukaryotic cells, higher order organized movements have been also reported for migrating cells confined in circular micropatterns (*Doxzen et al., 2013*; *Segerer et al., 2015*; *Wan et al., 2011*) or spheroids (*Chin et al., 2018*). For some of these systems, the connection between a macroscopic collective behavior and the microscopic dynamics of its constituents has been established. For the traveling band formation of biopolymers and molecular motors, local physical interactions among

**eLife digest** The cells of animals and many other living things are able to migrate together in groups. This collective cell migration plays crucial roles in many processes in animals such as forming organs and limbs, and healing wounds.

A soil-dwelling amoeba called *Dictyostelium discoideum* – or just Dicty for short – is commonly used as a model to study how groups of cells migrate collectively. Individual Dicty cells may live alone but sometimes many cells come together to form a larger mobile structure called a "slug". Chemical signals coordinate how the cells collectively migrate to form the multicellular slug. Mutant Dicty cells that lack these chemical signal processes can still move together as a band that travels across a surface. This movement resembles a type of collective motion that has previously been observed in physics experiments using self-propelled particles. However, it remains unclear how this collective behavior works.

Hayakawa et al. have now combined genetics, cell biology and computational approaches to study how groups of the mutant Dicty cells migrate together. The experiments showed that the traveling band is dynamically maintained by cells joining or leaving, and that this turnover is caused by simple interactions between the cells known as "contact following locomotion".

Contact following locomotion has been also reported in mammalian cells so the findings of Hayakawa et al. may aid research into how animals develop and how errors in cell migration may lead to diseases. Further studies are required to find out whether other cells showing contact following locomotion also travel in a band.

constituent elements effectively works as an alignment interaction, which induce the collective motion (*Schaller et al., 2010*; *Sumino et al., 2012*; *Suzuki et al., 2015*). In the case of eukaryotic cells, however, the connection between macroscopic traveling band formation (*Kuwayama and Ishida, 2013*) and microscopic short-range cell–cell interactions remains unclear. In particular, quantitative characterization of the traveling band formation and genetic analysis to reveal responsible cell–cell interaction have not been performed yet.

The social amoeba *Dictyostelium discoideum* is a model organism for the study of collective cell migration. The coordinated movement of cell population is achieved by individual chemotactic motion to the cAMP gradient, which is formed in a self-organized way. However, a mutant cell that lacks chemotactic activity to cAMP still exhibits an organized coordinated motion that is probably mediated by cell-cell contacts (*Kuwayama and Ishida, 2013*). Here, we demonstrate that this coordinated motion is a spontaneous polar order formation which phase-separates with a disordered background. We further show that this polar order formation is attributable to the tail-following behavior among the migrating cells, called contact following locomotion (CFL). We find that the polar ordered phase caused by CFL has an internal structure. An agent-based model with CFL further reveals that this internal structure is characteristic of the CFL-induced polar order formation. Thus, we establish the link between the collective behavior and the cell-cell interactions. Our findings open new possibilities that the concept of self-propelled particles contributes to the understanding of a highly orchestrated biological event of migrating cells in multicellular systems.

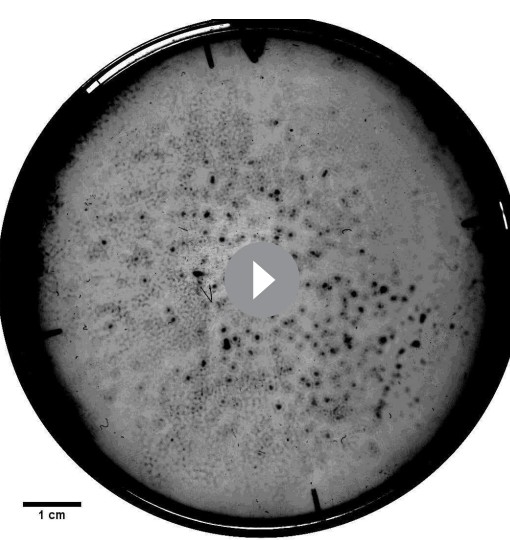

**Video 1.** Macroscopic observation of the propagating bands. The video was taken every 15 min for 28.5 hr. Video acceleration: 11400 × real time.
https://elifesciences.org/articles/53609#video1

## Results

### Traveling band formation of non-chemotactic *Dictyostelium* cells

In the present study, we investigated collective cellular motion in a mutant strain of *Dictyostelium discoideum*, known as "KI cell," which lacks all chemotactic activity (*Kuwayama and Ishida, 2013*; *Kuwayama et al., 1993*), and thus does not form a cell aggregate under starvation conditions. Wild-type *Dictyostelium discoideum* forms an aggregate as a result of chemotaxis mediated by a self-secreted extracellular chemoattractant. Under starvation conditions, KI cells spread on a non-nutrient agar plate show a segregation of cell density, which propagates as bands in around six hours (*Kuwayama and Ishida, 2013*), when the cell density is within a particular range ($1.0 \times 10^5$ cells cm$^{-2}$ to $4.0 \times 10^5$ cells cm$^{-2}$) (*Video 1*). Initially, the traveling bands propagate in random directions with high orientational persistence. When two bands collide, they appear to pass through each other, retaining their shapes (*Figure 1a* left) (*Kuwayama and Ishida, 2013*). However, over time, the propagation directions gradually become aligned, probably due to weak reorientation of propagation direction as an effect of collisions. Finally, the bands are arranged almost periodically in space with a spatial interval of about 1 mm (*Figure 1a right, b*).

To determine the mechanism underlying this collective cellular motion, we conducted high-magnification observations. At around 16 hours after cells were spread on an agar plate, a punched-out section of the agar plate was placed upside down on the glass slide, such that the monolayer of cells was sandwiched between agar and glass (*Figure 1—figure supplement 1a*). These cells formed a high-density area that moved as a band in low-density area for long periods of time with high orientational persistence (*Figure 1c, d* and *Video 2*). Whereas the cells in high-density area are packed without extra space, and thus the cell density is similar across different samples (*Figure 1—figure supplement 1c*), the size $W$ of the band along the propagation direction showed a broad distribution, ranging from $W = 200$ μm to 700 μm (N=10) (*Figure 1—figure supplement 1b*). In contrast, the traveling speed $\langle v_b \rangle = 0.5 \pm 0.03$ μm/s (N=10) was consistent among different bands, independent of size $W$ (*Figure 1—figure supplement 1b*).

### Analysis of single cell trajectories

To study the relationship between these collective behaviors and the migration of individual cells, we next performed cell-tracking analysis. Cellular movements were recorded by tracing the motion of fluorescent microbeads that were incorporated into the cells by phagocytosis. *Figure 2a* shows typical trajectories of individual KI cells. The distribution of migration speeds indicates that cell migration speed inside the band is slightly faster than that outside the band (*Figure 2b*). The average migration speeds of individual cells inside and outside the band were $v_{in} = 0.38 \pm 0.14$ μm s$^{-1}$ and $v_{out} = 0.30 \pm 0.16$ μm s$^{-1}$, respectively. The migration direction of the cells inside the band was distributed around the direction of band propagation (176.2 degrees, *Figure 2c*), although the fluctuation around the average direction is relatively large (standard deviation was 42.7 degrees). In contrast, the migration direction outside the band was distributed almost uniformly (*Figure 2c*). The mean squared displacement (MSD) inside the band was proportional to $t^2$ for more than $10^3$ s (*Figure 2d*). In contrast, the MSD outside the band exhibited a transition at around 100 s. from a persistent motion proportional to $t^2$, to a random motion proportional to $t$, which indicates that this motion can be described as a persistent random motion with no preferred direction (*Figure 2d*). This observed directional randomness reflects the effects of cellular collisions, as well as its intrinsic nature of single cells. In sum, cells inside the band exhibit directionally persistent motions, whereas cells outside move randomly.

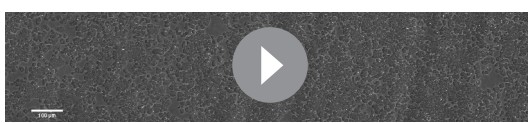

**Video 2.** Microscopic observation of the propagating bands. The video was taken every 15 s for 1.66 hr. Video acceleration: 230 × real time.
https://elifesciences.org/articles/53609#video2

### Propagation of cell density profile

We then compared the average cell speed inside the band $v_{in}$ and the band propagation speed $v_b$, (*Figure 2e*), and found that the band propagates faster than the cell migration speed for all samples investigated. This implies turnover of cells in the band, and that the band is continuously assembled at the front of the band and

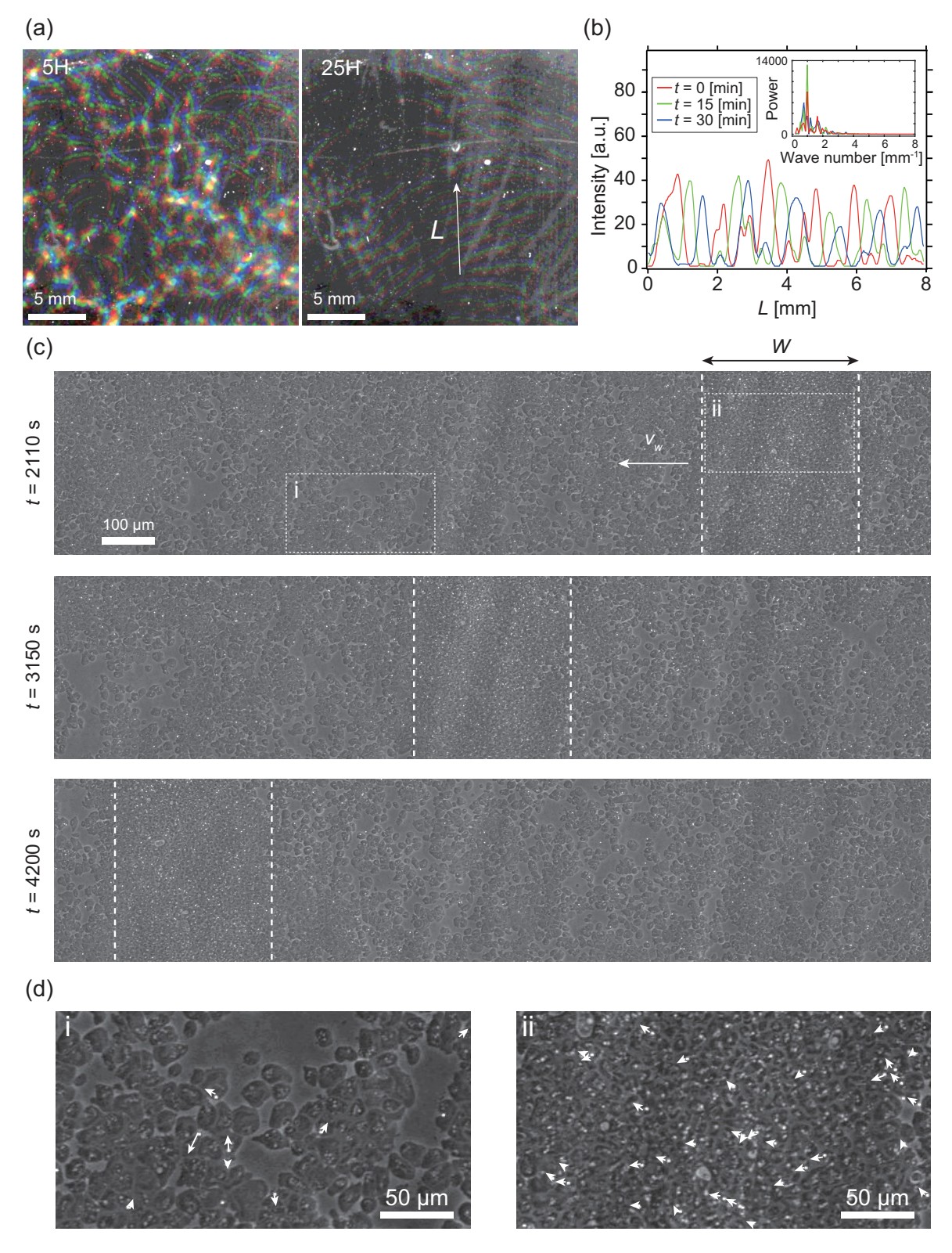

**Figure 1.** Segregation of cell density and formation of bands in non-chemotactic *D. discoideum* KI cell. (a) The density profile of three time points with a time interval of 15 min indicated by color-coding (red $t$ = 0 min, green 15 min, blue 30 min). Brighter color indicates higher density. Five (left) and 25 (right) hours after incubation. See also *Video 1*. (b) The intensity profile along the line indicated in (a), showing a periodic distribution of high-density regions. The inset shows a power spectrum of the intensity profile, indicating that the spatial interval was about 1 mm. (c) Time evolution of phase-

*Figure 1 continued on next page*

*Figure 1 continued*

contrast image of high-density region (dotted lines) at *t* = 2110, 3150, 4200 s, respectively. The time points correspond to that in *Video 2*. (**d**) High magnification images of low-density region (i) and high-density region (ii). Arrows indicate the migration directions of cells.

The online version of this article includes the following source data and figure supplement(s) for figure 1:

**Source data 1.** Statistics of traveling bands.
**Figure supplement 1.** Experimental setup and properties of traveling band.

disassembled at the back. Thus, it is the cell density profile that shows propagation as a band (*Kuwayama and Ishida, 2013*). Such turnover of cells is also evident from the individual trajectories shown in *Figure 2a and f*, where the trajectories started in a low-density region entered a band (high-density region) at its front, and then left the band from its back.

## Analysis of multicellular movement reveals polar order formation

To quantitatively characterize the multicellular movement, we introduce the local polar order parameter, $\varphi(n,t) = \left| \langle \boldsymbol{v}_i(t)/|\boldsymbol{v}_i(t)| \rangle_{i \in \mathfrak{L}(n)} \right|$, obtained from the instantaneous cell velocity $\boldsymbol{v}_i(t)$, where $\mathfrak{L}(n)$ is the $n$th domain along the direction of band propagation (see Materials and methods). In the high-density region that propagates as a band, $\varphi(n,t)$ reaches around 0.8, while $\varphi(n,t)$ in the low-density area remained below 0.4 (*Figure 3a*). Thus, the high-density region is polar-ordered phase, which propagates in the low-density disordered phase. The polar order parameter of the band showed intersample variability, and was distributed from 0.6 to 0.85 (*Figure 3b*). We found that the order parameter of band was positively correlated with the width of band $W$ (*Figure 3b*).

## Internal structure in the polar ordered region

The polar order phase is not completely homogeneous with respect to migration direction, but exhibits heterogeneity; this is related to the underlying assembly mechanism. This heterogeneity can be visualized in the velocity field obtained by optical flow, in which the direction of cell migration can be distinguished by color (*Figure 3c* and *Video 3*). The size-dependent squared local order parameter $\langle \varphi_l^2(s) \rangle$ (see Materials and methods) shows a logarithmic decay with area $S$ (*Figure 3e*), indicating that this heterogeneity is not spatially uncorrelated. Within the band (*Figure 3c* bottom), the migration direction was widely distributed from about 145 to 210 degrees (a black line in *Figure 3f*; the mean is 178.1 degrees and the standard deviation is 31.2 degrees). The probability density functions (pdf) of the migration direction obtained for the four regions (*Figure 3c* bottom (i–iv)) show peaks at different directions (*Figure 3f*), indicating the presence of two subpopulations; one in which the migration direction is ~160 degrees (regions (ii) and (iv)) and another in which it is ~190 degrees (regions (i) and (iii)). These two subpopulations are also recognized in *Figure 3c* (bottom) as the regions with dark blue and light green colors, respectively, forming stripes. These two types of stripes extend perpendicular to the direction of band propagation, and are alternately arranged. The typical width of the stripe was around 125 $\mu$m, as determined by the analysis of autocorrelation function (*Figure 3—figure supplement 1a*). The kymograph in *Figure 3d* shows the temporal evolution of the velocity field along the line PQ in *Figure 3c*, indicating that the stripes (light green and dark blue) are almost immobile, suggesting that the same cells experience the two stripes sequentially. In a reference frame co-moving with the band, cells move from the front to the end, since the speed of traveling band is faster than the speed of cells (*Figure 2e*, see also *Figure 3—figure supplement 1b*). During this relative motion of cells from the front to the end of a band, they move downward in a stripe, and then enter the next stripe moving upward direction (*Figure 3—figure supplement 1b*).

We also tested if the similar behavior can be seen in the single cell trajectories analyzed in *Figure 2*. As shown in *Figure 3—figure supplement 4*, although the density of tracked cell is quite sparse, we found that the migration directions indicated by colors also exhibit heterogeneity with light green and dark blue (*Figure 3—figure supplement 4a*). The temporal average of migration direction at a given *x* position indicates that the direction is distributed from about 150 to 210 degrees (*Figure 3—figure supplement 4b*), which is consistent with the optical flow analysis (*Figure 3f*). Furthermore, the trajectories move through regions with different colors, implying that

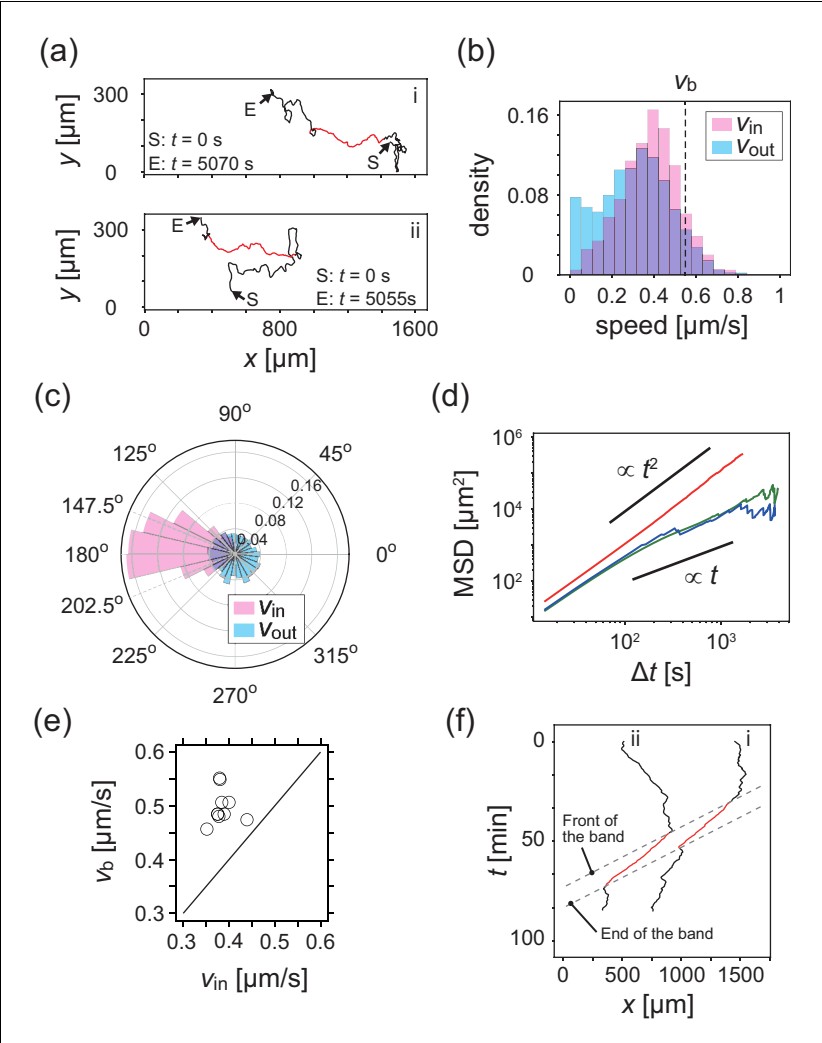

**Figure 2.** Analysis of single cell migrations inside and outside the band. (a) Trajectories of single cells inside (red) and outside (black) the band taken every 15 s. These trajectories were taken from the data shown in *Figure 1c*. (b, c) The distributions of the migration speed (b) and migration direction (c) inside (pink) and outside (blue) the band shown in *Figure 1c*. The number of trajectories analyzed is N=35. In *Figure 1c*, the average of migration direction inside the band (pink) was 176.2 degrees and the standard deviation was 42.7 degrees. (d) Mean squared displacement (MSD) of cell motions inside the band (red), before entering the band (green) and after leaving the band (blue) (e) Scatter plot of the band speed $v_w$ against the cell speed $\langle v_{in} \rangle$ within the band. The number of bands investigated is $N = 10$. Trajectory data and traveling band position are available in *Figure 2—source datas 1* and *2*. Source data for (e) is available in *Figure 1—source data 1*. (f) Spatiotemporal plot of trajectories shown in a. The x coordinates of trajectories (horizontal axis) are plotted against time (vertical axis). The front and back of traveling band are shown by dotted lines.

The online version of this article includes the following source data for figure 2:

**Source data 1.** Trajectory data of 35 cells in the experiment shown in *Figure 1c*.

**Source data 2.** Front and end positions of band as functions of time in the experiment shown in *Figure 1c*.

cells change their migration direction following the flow direction in the regions (*Figure 3—figure supplement 4a*). From the trajectories used in *Figure 3—figure supplement 4a*, the cell migration speed in the x-direction was $0.28 \pm 0.09~\mu m/s$ , while the width of stripes was estimated to be around 50 to 100 $\mu m$. Thus, the time scale that cells pass across a stripe is about 200-400 s, if the stripe is almost immobile as we have shown above. The time interval that cells travel across the traveling band is obtained as 1120 s. Thus, cells pass several different stripes during the time interval that cells

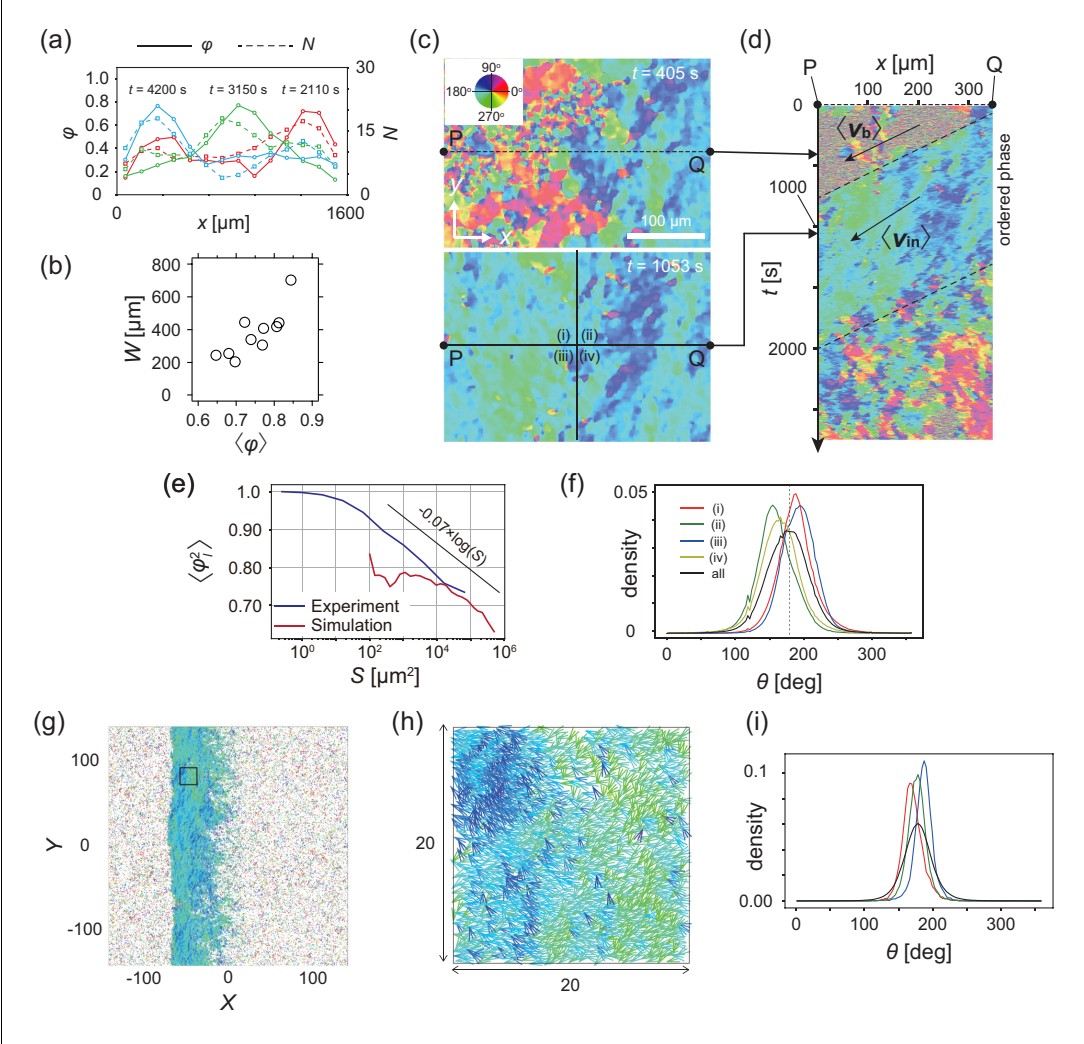

**Figure 3.** Analysis of heterogeneity within the ordered phase. (**a**) Spatial profile of the local polar order parameter (solid lines) and the number of beads in the intervals (dotted lines) in *Figure 1c*. (**b**) Scatter plot of band width against polar order parameter within the band region. The number of bands studied is $N = 10$. (**c**) Optical flow images in the front region of a band $t=405$ (top) and within the band $t=1083$ (bottom). See also *Video 3*. (**d**) Kymograph of the optical flow image along the line PQ shown in (**c**) (top). The arrows indicate the average velocity of band $\langle v_b \rangle$ and the average cell speed $\langle v_{in} \rangle$. (**e**) Size-dependent squared local order parameter plotted against the area $S$ for the data shown in (**c**). (**f**) Probability distribution function (pdf) of the migration direction within the band region obtained by averaging the pdfs of the sequential 150 frames in *Video 3*. The pdfs (i)-(iv) are obtained in the regions (i)-(iv) in (**c**) (bottom), respectively. Average pdf is shown by the black line (the mean is 178.1 degrees and the standard deviation is 31.2 degrees). (**g**) Snapshot of simulation result showing a polar ordered phase as a propagating band in the background of disordered phase. The color code indicates the migration direction of individual particle as shown in (**c**). See also *Video 8*. (**h**) Magnification of squared area shown in **g**. The size of area (20x20) is comparable to the whole area shown in (**c**). Each arrow indicates the direction of polarity. (**i**) Probability distribution function of the migration direction within the band region in the simulation (red, green and blue lines). For the choice of ROI, see Materials and methods. Average pdf is shown by the black line. Source data for (**b**) is available in *Figure 1—source data 1*.

The online version of this article includes the following figure supplement(s) for figure 3:

**Figure supplement 1.** Internal structure within a traveling band.

**Figure supplement 2.** Phase diagram of the agent-based simulation for changes of CFL $\alpha$ and the alignment effect $\zeta$.

**Figure supplement 3.** Analyses of heterogeneityin the numerical simulations.

**Figure supplement 4.** Analysis of internal structure from individual cell trajectories.

stay in the traveling band. These analyses illustrate that the polar order phase possesses an internal structure with respect to the migration direction.

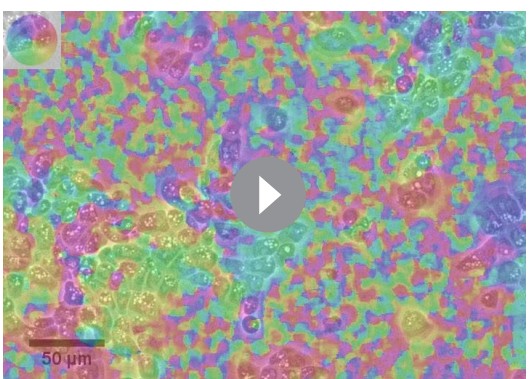

**Video 3.** Propagating band with overlaying the coloring based on the optical flow analysis. The video was taken every 3 s for 39.5 min. Video acceleration: 151 × real time.

https://elifesciences.org/articles/53609#video3

## Contact following locomotion is the cell-cell interaction that induces polar pattern formation

The formation of a polar-ordered phase with an internal structure is ultimately related to the microscopic interactions between individual cells, which are short-ranged. In the low-density region, cells are not completely isolated, but rather are often associated with each other, migrating in single files (*Figure 4a* and *Video 4*). This tail-following behavior has been described for wild-type *Dictyostelium* cells within aggregation streams (*Dormann et al., 2002*). We call this behavior 'contact following locomotion' (CFL). In low-density assay, when two cells collide, they either form CFL (*Figure 4b* and *Video 5*) or not (*Figure 4—figure supplement 1a*). To quantitatively characterize CFL, we measured the duration of cell–cell contact after two cells collide. During the formation of CFL, the typical cell-to-cell distance is given by $d_a = 24\ \mu\mathrm{m}$. We measured the time interval during which the distance is less than $d_a$ from the time series of the distance between two cells (*Figure 4—figure supplement 1b*). As shown in *Figure 4c*, in half of the cases, cell–cell contact persists for more than 300 s. To determine whether cells that form contacts for >300 s exhibit CFL or side-by-side behavior, we measured the average angle $a$ of the angles $a_1$ and $a_2$, which are the angles of the velocity vectors $v_1$ and $v_2$ with respect to the vector connecting the two cell centers $d$, respectively (*Figure 4d*). In almost 60% of all cases, the angle $a$ is 0–30 degrees (*Figure 4e*) that corresponds to CFL.

To determine whether CFL is responsible for the collective behavior of KI cells, we sought a mutant cell that lacks CFL activity. A knockout mutant that fails to express the cell–cell adhesion molecule TgrB1 exhibits reduced CFL activity (*Fujimori et al., 2019*). TgrB1 is known to mediate cell–cell adhesion via a heterophilic interaction with its partner TgrC1 (*Hirose et al., 2011*; *Hirose et al., 2015*; *Li et al., 2015*; *Fujimori et al., 2019*). We first assessed whether the *tgrB1* null mutant forms propagating bands. As in the control case, under starvation conditions, we spread the *tgrB1* null mutant cells on a non-nutrient agar plate at a cell density of $2.0$ to $3.0 \times 10^5$ cells cm$^{-2}$ (see Materials and methods). However, neither segregation of cell density nor propagating bands appeared (*Videos 6* and *7*).

We then quantitatively characterized the formation of cell–cell contacts. We found that in 80% of all cases, cell–cell contact is disrupted before 300 s (*Figure 4f*), and that only 10% of cells established CFL (*Figure 4—figure supplement 1d*). In particular, in half of all cases, the cell–cell distance becomes larger than $d_a$ in 120 s, indicating that these cells failed to establish cell–cell contact. Thus, our analyses illustrate that in the *tgrb1* null mutant, CFL is nearly absent. Since TgrB1 is a protein that can mediate cell-cell adhesion as well as contact-dependent signaling, it is reasonable that the *tgrB1* null mutant cell does not show CFL. This analysis suggests that the reduced ability to perform CFL can be linked to the defect of *tgrB1* mutant in the formation of

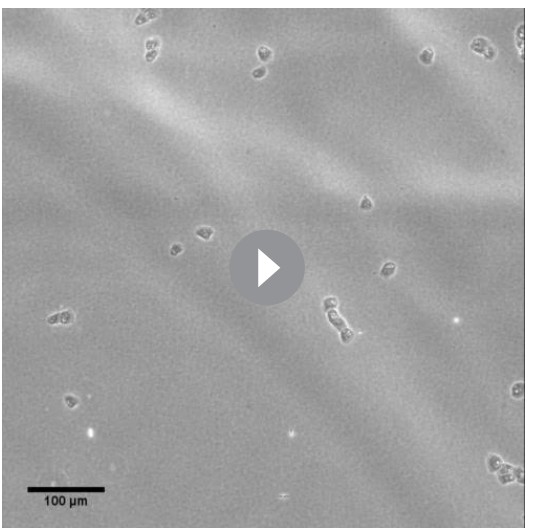

**Video 4.** Migration of the KI cells in the low-density region. The video was taken every 15 s for 2 hr. Video acceleration: 378 × real time.

https://elifesciences.org/articles/53609#video4

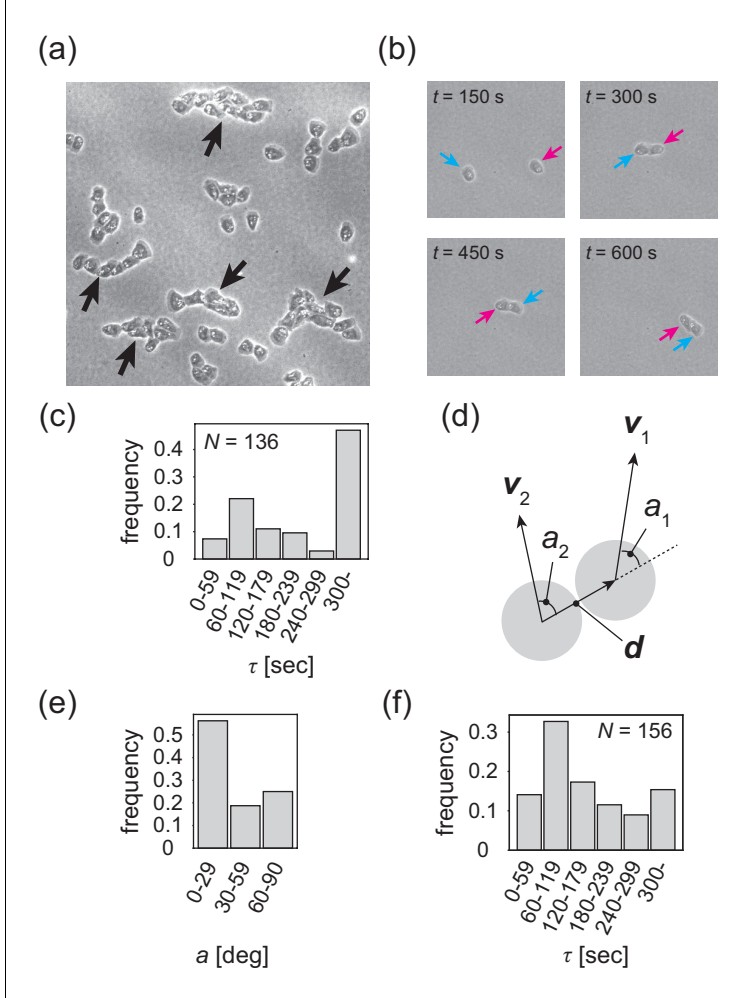

**Figure 4.** Contact following locomotion responsible for band propagation. (**a**) Snapshot of the contact following locomotion. See also *Video 4*. (**b**) Representative time evolution of collision of two cells. Colored arrows represent the same cell. See also Video 5. (**c**) Histogram of the duration of two cell contacts for KI cell (control). (**d**) Schematic of angle $a_1$ ($a_2$), which is the angle of the velocity vector $v_1$ ($v_2$) with respect to the vector $d$ connecting two cell centers. Then, the angle $a$ is obtained as the angular average of $a_1$ and $a_2$, that is $A\cos a = (\cos a_1 + \cos a_2)/2, A\sin a = (\sin a_1 + \sin a_2)/2$. (**e**) Histogram of the angle $a$ for the KI cells that contact each other for >300 s. (**f**) Histogram of the duration of two cell contacts for the *tgrb1* null mutant. Source data for (**c**) and (**f**) are available in *Figure 4—source datas 1* and *2*.

The online version of this article includes the following source data and figure supplement(s) for figure 4:

**Source data 1.** Source data of duration of two cell contacts for KI cell shown in *Figure 4c*.
**Source data 2.** Source data duration of two cell contacts for *tgrb1* mutant cell shown in *Figure 4f*.
**Figure supplement 1.** Collision analysis and locomotive activity of KI and *tgrB1* mutant cells.

traveling band formation, although we cannot exclude other effect that could explain the phenotype of this mutant. For instance, if there are some changes in the locomotive activity of individual cell due to the mutation of *tgrB1*, it could also affect the formation of traveling band. Thus, we next compared the locomotive activities between control cells and *tgrB1* null mutants. The velocity auto-correlation functions $C(\Delta t)$ of the isolated single cells showed similar behaviors (*Figure 4—figure supplement 1c*), indicating that locomotive activities were comparable between KI cells and the *tgrB1* null mutant cells. The above analyses suggest that the difference in the cellular scale behavior between control KI cell and *tgrb1* mutant cell is the ability of CFL, and we thus conclude that CFL is essential for the segregation of cell density and the formation of propagating bands.

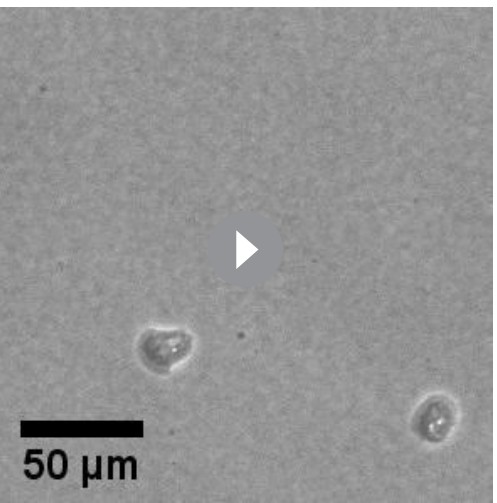

**Video 5.** A binary collision of the KI cells in the low-density assay. The video was cropped from the video of the low-density assay with a length of 10.25 min. Video acceleration: 153 × real time.
https://elifesciences.org/articles/53609#video5

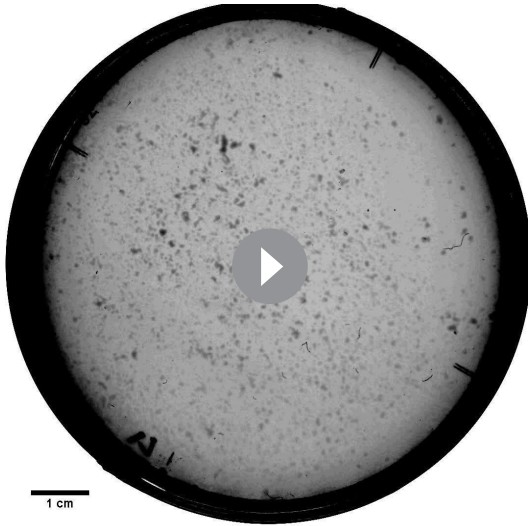

**Video 6.** Macroscopic observation of the population of tgrb1 null mutant. The video was taken every 15 min for 28.5 hr. Video acceleration: 11400 × real time.
https://elifesciences.org/articles/53609#video6

## Mathematical modeling of polar pattern formation driven by contact following locomotion

The collective motion of KI cells induced by the CFL interaction can be modeled by an agent-based simulation (*Hiraiwa, 2019*). In the model, particle $i$ at position $\mathbf{r}_i$ self-propels at a constant velocity $v_0$ in the direction of its own polarity $\mathbf{q}_i$ subjected to white Gaussian noise. Thus, without interactions, the particles exhibit a persistent random walk (*Hiraiwa et al., 2014*). The effect of CFL is introduced so that polarity $\mathbf{q}_i$ orients to the location of the adjacent particle $j$, when particle $i$ is located at the tail of particle $j$ (parameterized by $\zeta$). In addition to this effect, the particles interact with each other through volume exclusion interaction, adhesion, and the effect of polarity $\mathbf{q}_i$ orienting toward the direction of its velocity $\mathbf{v}_i = d\mathbf{r}_i/dt$ (parameterized by $\alpha$). For a fixed parameter set ($\alpha = 0.4$; see Materials and methods), without CFL ($\zeta = 0$), the collective behaviors did not form (*Figure 3—figure supplement 2*). In contrast, with CFL ($\zeta \geq 0.1$), a polar-ordered phase appeared as a propagating band in the background of disordered phase (*Figure 3g* and *Video 8*). The speeds of the traveling band and particles within the band were 0.96 and 0.9, respectively, relative to the speed of isolated particles, indicating that the band is dynamic with assembly in the front and disassembly in the tail, consistent with our experimental results. From the spatial pattern shown in *Figure 3gh*, in which the migration direction is indicated by color code, heterogeneity in the migration direction is recognized within the polar-ordered phase. In the simulation, we studied the pdf of migration direction in regions, whose size is comparable to that in *Figure 3c* ((i)–(iv)), and found that the pdf exhibited peaks at different directions (*Figure 3i*), similar to our experimental results (*Figure 3f*). To determine whether this formation of internal structure is a characteristic of propagating bands induced by CFL, we studied a propagating band formed by increasing alignment effect $\alpha$ without CFL ($\zeta = 0$), and found that the pdfs of migration direction exhibit peaks at closely similar positions, indicating that the migration direction in the ordered phase is more homogeneous (*Figure 3—figure supplement 3*). Thus, the formation of internal structure appears to be a characteristic of the collective behavior induced by CFL. The size-dependent squared local order parameter $\langle \varphi_l^2(s) \rangle$ (see Materials and methods) also shows the characteristic decay with a logarithmic dependence on area $S$ (*Figure 3e*), as observed experimentally.

## Discussion

In this study, we report that a mutant of *Dictyostelium* cell that lacks all chemotactic activity exhibits spontaneous segregation into polar ordered solitary band (*Kuwayama and Ishida, 2013*). This

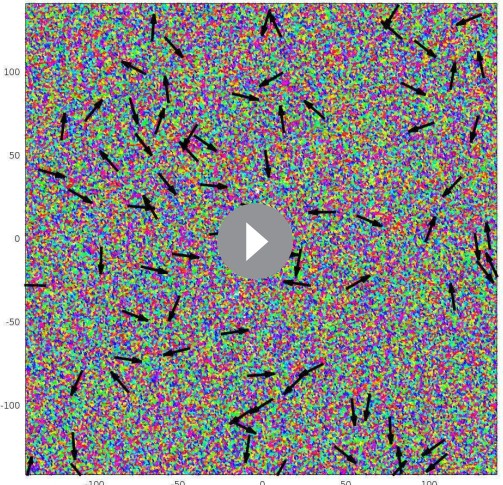

**Video 7.** Microscopic observation of the population of tgrb1 null mutant. The video was taken every 15 s for 1.25 hr. Video acceleration: 225 × real time.
https://elifesciences.org/articles/53609#video7

pattern formation is attributable to the cell-cell interaction called contact following locomotion (CFL) (*Figure 4*). The agent-based model that includes CFL reproduces the observed macroscopic behaviors (*Figure 3g*). Thus, we establish a link between the microscopic cell-cell interactions and the macroscopic polar pattern formation.

We showed that the width of band is distributed widely from $W = 200$ μm to 700 μm (*Figure 1—figure supplement 1b*), and found the positive correlation between the width and the order parameter within the band (*Figure 3b*). The local cell density within the band is similar across different samples (*Figure 1—figure supplement 1c*), suggesting that the local cell density may not be a relevant factor for the increase in the order parameter. We speculate that if the correlation in the migration direction is gradually decorrelated from the front to the end of the band, bands with lower order parameters will be more prone to larger decorrelation in the migration direction. Consequently, we expect that the stronger the polar order, the wider the band width $W$.

One characteristic behavior of the present polar pattern formation is the formation of internal structure, which consists of subpopulations with transversal motions (*Figure 3c, d, f*. From the numerical simulation result, this formation of subpopulation was not seen in the model without CFL (*Figure 3—figure supplement 3d, e*). Thus, the internal structure is a characteristic of CFL induced polar pattern formation. A population of cells enters the band at its front with directional alignment induced by CFL in random direction. During the relative movement of these cells from the front to the end of band, the migration direction may not be dampened completely to the direction of band propagation probably due to the directional persistence induced by CFL. In this way, subpopulations with respect to the migration direction are formed when CFL is present. A full analysis of this mechanism remains to be a future topic.

In this paper, we mainly focused on the behavior of single solitary band. We studied the traveling band, which was well separated from other bands. Thus, all properties of single solitary band studied in this paper is independent of interaction between different bands. In some area, the traveling bands are arranged almost periodically in space with a spatial interval of about 1 mm (*Figure 1b*). How bands interact with each other to reach a periodic spacing and whether the interval is independent of band width $W$ are to be investigated.

Wildtype *Dictyostelium discoideum* usually aggregates through chemotaxis to form a hemispherical mound with a central tip region that regulates the formation of slug-like multicellular structure (*Williams, 2010*). It has been suggested, however, that other mechanism also involves in the formation of aggregate, such as contact following (*Dormann et al., 2002*; *Fujimori et al., 2019*; *Shaffer, 1962*; *Umeda and Inouye, 2002*). In fact, whereas the KI cell alone does not form the multicellular structure, KI cells are able to spontaneously migrate to the central tip region transplanted from a wildtype slug and undergo normal morphogenesis and cell differentiation; this is not observed in mutant KI cells lacking TgrB1 (*Kida et al., 2019*), suggesting that TgrB1-dependent CFL without chemotaxis allows KI cells to spontaneously migrate in slug. Furthermore, in wildtype cells, the chemical guidance cue has been shown to cease during the

**Video 8.** Propagating band formation generated in the agent-based simulation. The color code indicates the migration direction of individual particle as shown in *Figure 3c*. Arrows indicate the direction of polarity.
https://elifesciences.org/articles/53609#video8

multicellular phase, which suggests that an alternative mechanism induces collective cell migration in the multicellular body (*Hashimura et al., 2019*). We propose that polar order formation induced by CFL plays an important role in late-stage morphogenesis in this organism. Contact following locomotion, or chain migration, have been reported in other cell types (*Li and Wang, 2018*). The macroscopic behaviors reported in this paper may thus be found in other systems as well.

# Materials and methods

## Key resources table

| Reagent type (species) or resource | Designation | Source or reference | Identifiers | Additional information |
|---|---|---|---|---|
| Cell line (*Dictyostelium discoideum*) | KI-5 | National BioResource Project Cellular slime molds | NBRP ID: S00058 | Available in National BioResource Project Cellular slime molds (https://nenkin.nbrp.jp) |

## Culture condition of KI mutant cells and cell density measurement

1 mL of *Klebsiella aerogenes* suspended in 5LP medium (0.5% Lactose, 0.5% bactopeptone 211677, Optical density = 0.1) was spread on the 9 cm 5LP plate (0.5% Lactose, 0.5% bactopeptone 211677, 1.5% agar), 5LP medium dried, the non-chemotactic *Dictyostelium discoideum*, KI mutant cells were inoculated on the plate. The KI cells were incubated for about five days at 21°C. After cultivation, the KI cells and *Klebsiella* on the plate were collected with a phosphate buffer (PB). To remove the *Klebsiella*, the suspension was centrifuged and discard as much of the supernatant liquid as possible by aspiration, then clean PB was added. After repeating this process two times, the number of cells was counted using a hemacytometer.

## Macroscopic observation of the traveling bands

The washed KI cells were spread (cell density = $5.0 \times 10^5$ cells/cm$^2$) on a 9 cm non-nutrient agar plate (1.5% agar) to cause starvation. After drying of the PB, the plate was scanned every 15 min using a film scanner (V850, EPSON). The brightness in scanner images is inversely correlated with cell density (*Takeuchi et al., 2014*). For *Figure 1a and b*, the original images were inverted with color that depends on time points.

## Microscopic observation of the traveling bands

The KI cells were spread (cell density = 2.0 to $3.0 \times 10^5$ cells/cm$^2$) on the non-nutrient agar plate and incubated at 21°C for around 16 hr. A punched-out piece of the agar plate was placed upside down on the glass slide, and the travelling bands between the agar and glass was observed by phase contrast imaging. For *Figures 1* and *2*, *3a, b* and *4*, the images are taken every 15 s, using an inverted microscope (TiE, Nikon, Tokyo Japan) equipped with camera (iXon+, Andor Technology) and a 20x phase-contrast objective. For *Figure 3c–f*, the images are taken every 3 s, using an inverted microscope (TiE, Nikon, Tokyo Japan) equipped with camera (DS-Fi3, Nikon, Tokyo Japan) and a 20x phase-contrast objective.

## Tracking analysis of individual KI cells

For the tracking analysis shown in *Figure 2* and *Figure 3* and 1 µL of the PB including 3% fluorescent microbeads (ex:441, em:486, 1.0 µm, Polysciences, Inc) was spread at the same time with the KI cells. The trajectories of the microbeads were automatically tracked by using the ParticleTracker 2D, a plugin for Image J (National Institutes of Health, USA). To eliminate the trajectories of the microbeads that was not internalized by the KI cells, if $|v_{cell}|$ was slower than 0.25 µm/s for 300 s continuously, we excluded such trajectories.

For the analysis shown in *Figure 2*, we used the trajectories longer than 1 hr. The number of trajectories analyzed in *Figure 2b–d* was N = 35 with 2044 time points for the cells inside of the band and 9095 time points for the cells outside of the band. For *Figure 2e*, we performed the same experiment for 10 times. The numbers of trajectories that last for more than 1 hr in the 10 samples were N = 35, 19, 34, 12, 13, 17, 19, 16, 18, 7.

## The mean squared displacement (MSD)

The MSD (**Figure 2d**) was calculated using the formula below.

$$\text{MSD(t)} = \frac{1}{N(T-t)}\sum_{i=1}^{N}\sum_{t}^{T-t}\{r_i(t+t)-r_i(t)\}^2,$$

where $\Delta t$, $T$, and $N$ means a time interval, final time, and number of the trajectory, respectively.

## Polar order parameter

To obtain the local polar order parameter $\varphi(n,t)$ shown in **Figure 3a**, the picture shown in **Figure 1c** was divided into $n$ sections with width $\Delta x$ (µm), and the order parameter was calculated in each section at each time from the trajectories obtained by the tracking analysis. The local order parameter $\varphi$ is defined as

$$\varphi(n,t) = \left| \frac{1}{N(n)}\sum_{i \in \mathfrak{L}(n)} \frac{v_i(t)}{|v_i(t)|} \right|,$$

where $\mathfrak{L}(n)$ is the set of cells that satisfy $(n-1)\Delta x \leq x_i \leq n\Delta x$, $N(n)$ is number of the cells in $\mathscr{L}(n)$, $v_i$ and $x_i$ are the velocity and $x$-position of $i$-th fluorescent microbeads, respectively. In this study, $n = 14$ and $\Delta x = 119$ µm.

To obtain the order parameter $\langle \varphi \rangle$ of the traveling band used in **Figure 3b**, we first obtained $\varphi$ in the band region at each time step. Then, we took average it over all time steps.

## Optical flow analysis

Optical flow analysis was performed based on the Gunnar-Farneback method using OpenCV library. In the optical flow analysis, the displacement of each pixel in the original pictures are characterized by coloring based on the HSV ('hue', 'saturation', 'value') representation. The 'hue' varies depending on angular variation of each pixel. In this study, a 'saturation' and 'value' of the processed images via optical flow was fixed to 150 and 255, respectively. The sequential images of the traveling band used for this analysis were taken every 3 s.

## Size-dependent squared local order parameter

To characterize the internal structures of the traveling band, the size-dependent squared local order parameter $\langle \varphi_l^2(S) \rangle$ is introduced (**Figure 3e**). To obtain the size-dependent squared local order parameter, we first calculate the squared polar order parameter $\varphi_l^2(S)$ within a ROI of size $S$, which is defined as

$$\varphi_l^2(S) = \frac{1}{S^2}\left\{ \left(\sum_{(x,y)\in ROI} \cos\Theta(x,y)\right)^2 + \left(\sum_{(x,y)\in ROI} \sin\Theta(x,y)\right)^2 \right\}$$

where $\Theta(x,y) = \text{hue} \times (360/255)$ indicates the angular variation of the pixel at position $(x,y)$. The value of hue was obtained from the optical flow analysis (**Figure 3c**). Then, $\varphi_l^2(S)$ is averaged over the entire area to obtain $\langle \varphi_l^2(S) \rangle$. If $\Theta(x,y)$ is a random number without spatial correlation, as the increase of the area $S$, $\langle \varphi_l^2(S) \rangle$ is expected to decay in proportion to $S^{-1}$.

## The plot of the temporal average of the migrating direction in the band

Firstly, we divided the $x$-$t$ plane into the lattice with the interval of 20 µm (for $x$ axis) and 60 s (for $t$ axis). We then collect trajectories that pass through each lattice. In each lattice, the angle of migration direction was averaged. The obtained average angle in each lattice is shown with the color indicated in the color bar. Then, the traveling band region in $x$-$t$ plane was divided as shown in **Figure 3—figure supplement 4b** right. The temporal average of migration direction was taken at a given $x$ position, which was plotted in **Figure 3—figure supplement 4b** left.

## Autocorrelation function of transverse motion with respect to the band propagation direction

Because the band show propagation in *x*-direction, autocorrelation function of transverse motion $C_{\sin}$ is defined using *y*-component of motion as

$$C_{\sin}(x) = \frac{1}{Y(X-x)} \sum_{y=1}^{Y} \sum_{x=1}^{X-x} \sin\Theta(x,y)\sin\Theta(x+x,y),$$

where *x* is pixel interval along the *x*-axis. $C_{\sin}$ was plotted after that unit of *x* is converted to the length.

## Preparation of *tgrB1* null mutant cells

The gene disruption construct for *tgrB1* was synthesized by a polymerase chain reaction (PCR)-dependent technique (*Kuwayama, 2002*). Briefly, the 5-flanking region of the construct was amplified with two primers, 5-CAACAGGTGGAGACTTCGGG-3 and 5- GTAATCATGGTCATAGCTG TTTCCTGCAGGCCAGCAGTAATAGTTGGAG-3. The 3-flanking region of the construct was amplified with primers, 5- CACTGGCCGTCGTTTTACAACGTCGACGAGAACTGTTGATTCTGATGG-3 and 5- CTTGGTCCTGAACGAACTCC-3. The bsr cassette in the multicloning site of pUCBsr Bam (*Adachi et al., 1994*) was amplified using the primer pair 5-CTGCAGGAAACAGCTATGACCATGA TTAC-3 and 5-GTCGACGTTGTAAAACGACGGCCAGTG-3, both of which are complementary to the two underlined regions, respectively. The three amplified fragments were subjected to fusion PCR that produced the required gene-targeting construct. The gene-targeting constructs were cloned using a TOPO TA cloning kit for sequencing (ThermoFisher Scintific MA, USA). The linear construct was amplified by PCR using the outermost primers up to 10 µg and transformed into KI-5 cells. The KO clones were selected by genomic PCR using the outermost primers. *tgrB1* KO KI-5 cell (NBRP ID: S90519) is available in National BioResource Project Cellular slime molds (https://nenkin.nbrp.jp).

## Culture condition and starvation treatment of *tgrb1* mutant null cells

The *tgrb1* null cells were cultured in HL5 medium (1.43% Proteose Peptone 211684, 0.72% Yeast Extract212750, 1.43% Gulcose, 0.05% $KH_2PO_4$, 0.13% $Na_2HPO_4 12H_2O$) at 21 degrees Celsius. After reaching confluent, cells on the bottom were peeled off and collected, then washed two times with a centrifuge and PB. Next, the *tgrb1* null cells were transferred on the 1/3 SM plate (0.33% Gulcose, 0.33% bactopeptone 211677, 0.45% $KH_2PO_4$, 0.3% $Na_2HPO_4$, 1.5% agar) with *Klebsiella* suspension, and incubated for around two days at 21℃. After, through the wash and count, the *tgrb1* null cells were spread on the non-nutrient agar plate, after which the plate was scanned every 15 min using the film scanner.

## Characterization of the contact following locomotion

The KI cells and *tgrb1* null cells for the collision assay were scraped from the traveling bands and surface of the plate, respectively. The scraped cells were placed on the non-nutrient agar and sandwiched with the glass. After around one hour incubation at 21℃, binary collisions of two cells were observed by microscopy and recorded every 15 s. The motion of the cells was tracked manually using the Manual Tracking, a plugin of Image J. Here, collision was defined as the contact of pseudopods. We collected the data from three and four independent experiments for the KI cells and the *tgrb1* null cells, respectively. The total numbers of collision events are 136 (KI cells), and 156 (*tgrb1* null cells).

## The velocity autocorrelation function

Firstly, the migrations of the KI and *tgrb1* null mutant cells were recorded every 20 s for 60 min. Here, to extract an intrinsic locomotive activity of the cells, interactions with other cells, wall, and etc. were eliminated. Using obtained trajectories of cells that migrate with the velocity ***v***, the velocity autocorrelation function $C(\tau)$ was calculated. $C(\tau)$ is described with the form of

$$C(\Delta t) = \frac{1}{N(T-\tau)} \sum_{i=1}^{N} \sum_{t}^{T-\tau} \{\boldsymbol{v}_i(t+\tau) - \boldsymbol{v}_i(t)\}^2,$$

where $\tau$, $t$, $T$, and $N$ means a time interval, time, final time, and number of the trajectory, respectively (*Figure 4—figure supplement 1c*).

## Modeling collective motion induced by contact following locomotion

The collective motion of KI cells induced by the CFL interaction can be modeled by an agent-based simulation. In the model, self-propelled particle $i$ at position $\mathbf{r}_i$ moves at a constant velocity $v_0$ in the direction of its own polarity $\mathbf{q}_i$ subjected to white Gaussian noise. Thus, without interactions, the particles exhibit persistent random walk (*Hiraiwa et al., 2014*). Collective motion can be modeled by assuming particle-particle interactions (*Hiraiwa, 2019*). We firstly assume that the particles interact with each other through volume exclusion (parameterized by $\beta$) and adhesion (parameterized by $\gamma$). We also assume the feature that the polarity of each particle orients to the direction of its velocity $\mathbf{v}_i = d\mathbf{r}_i/dt$ (parameterized by $\alpha$); it is known that this assumption can effectively give rise to the alignment interaction between the particles when it is combined with the volume exclusion effect (*Li and Sun, 2014*). (Therefore, we simply refer to this feature as alignment effect in the main text.) As the main focus of this article, we incorporate CFL into this model by assuming the particle-particle interaction by which polarity $\mathbf{q}_i$ orients to the location of the adjacent particle $j$ when particle $i$ is located at the tail of particle $j$ (parameterized by $\zeta$). The equation of motion for the particle $i$ are then given by

$$\frac{d\mathbf{r}_i}{dt} = v_0 \frac{\mathbf{q}_i}{|\mathbf{q}_i|} - \beta \sum_{j \in \mathfrak{N}(i)} R \frac{\mathbf{r}_j - \mathbf{r}_i}{|\mathbf{r}_j - \mathbf{r}_i|^2} + \gamma \sum_{j \in \mathfrak{N}(i)} \frac{\mathbf{r}_j - \mathbf{r}_i}{|r_j - r_i|} \tag{1}$$

$$\frac{d\mathbf{q}_i}{dt} = I\mathbf{q}_i \left(1 - |\mathbf{q}_i|^2\right) + \mathbf{C}_i + \alpha \frac{\mathbf{v}_i}{|\mathbf{v}_i|} + \xi_i \tag{2}$$

where the second and third terms on the right-hand side of *Equation 1* are the effects of volume exclusion and adhesions, respectively. Here, $\mathfrak{N}(i)$ is a set of particles that are contacting with the particle $i$, that is the particle $j \in \mathfrak{N}(i)$ satisfies $|\mathbf{r}_j - \mathbf{r}_i| \leq R$. On the right hand side of *Equation 2*, the first term shows the self-polrization, the third term gives the effect that the polarity orients to the velocity direction $\mathbf{v}_i/|\mathbf{v}_i|$, the last term is white Gaussian noise with $\langle \xi_i \rangle = (0,0)$ and $\langle \xi_i(t) \cdot \xi_j(t') \rangle = \sigma^2 \delta_{ij} \delta(t - t')$, and the second term $\mathbf{C}_i$ describes the CFL, parameterized by $\zeta$, given by

$$\mathbf{C}_i = \frac{\zeta}{2} \sum_{j \in \mathfrak{N}(i)} \frac{\mathbf{r}_j - \mathbf{r}_i}{|\mathbf{r}_j - \mathbf{r}_i|} \left(1 + \frac{\mathbf{q}_i}{|\mathbf{q}_i|} \cdot \frac{\mathbf{r}_j - \mathbf{r}_i}{|\mathbf{r}_j - \mathbf{r}_i|}\right). \tag{3}$$

Here, when the polarity of particle $j$, $\mathbf{q}_j$, and the vector from particles $i$ to $j$, $\mathbf{r}_j - \mathbf{r}_i$, are in the same orientation, the maximum following effect is exerted on particle $i$ to the direction of particle $j$. Such a situation is expected when particle $i$ is located in the tail of particle $j$ with respect to the polarity $\mathbf{q}_j$. In contrast, when particle $i$ is located in the front of particle $j$, $\mathbf{C}_i$ almost vanishes. The simulation is implemented within a square box of size $L$ with periodic boundary condition. For all simulations, we used fixed parameter values except $\zeta$ and $\alpha$, given by $v_0 = 1.0, \beta = 1.0, R = 1.0, \gamma = 1.20$, and $\sigma^2 = 0.4$. For $I$ in *Equation 2*, we consider the situation where $I$ is infinitely large, so that $q$ was projected onto the unit vector $|q| = 1$ for the numerical simulation. The density of particles per unit area $\rho$ is given $\rho = 1$. The number of particles $n$ is $n = 80,000$ (*Figure 3g-i* and *Figure 3—figure supplement 3*) and $n = 10,000$ (*Figure 3—figure supplement 2*).

## Histogram of migration direction in the numerical results

Firstly, we selected only the ROIs in the vicinity of the band front in the following way: We define a ROI as being within the bands if the particle density is higher than 1.16, which corresponds to the 2D dense packing fraction of disks, ~0.91. Using this definition, we define the ROI as being vicinity of the band front if the ROI is within the band at the last $F_{ana}$ frames whereas it is out of the band at the frames between the last $F_{ana} + F_{wait} + F_{out}$ and the last $F_{ana} + F_{wait}$. In other words, $F_{ana}$ means the number of frames to be analyzed and must be within the band, $F_{out}$ means the number of frames to determine the band front (i.e. the frames in which the ROI must be still out of the band assuming that the band travels only in one direction), and $F_{wait}$ means the number of the waiting frames (i.e. the frames which are not used at all) between these frame sets.

The results of this algorithm for CFL-induced $(\zeta = 0.1, \ \alpha = 0.4)$ and alignment-induced $(\zeta = 0.0, \ \alpha = 1.0)$ bands are shown in *Figure 3—figure supplement 3a and b*, respectively. Here, we used the following sets of the parameters for our analysis in this article: $F_{ana} = 55$, which corresponds to the time window in the analysis of experimental data. $F_{wait} = 76$, with around which the band front can propagate across one ROI. $F_{out} = 5$, which has been empirically determined. The duration between each frame is $dt = 0.2$ in the unit of time of our numerical simulation.

Secondly, using these near-front ROIs, we calculate the histograms of migration direction $(\mathrm{d}\mathbf{r}_i(t)/\mathrm{d}t)/|\mathrm{d}\mathbf{r}_i(t)/\mathrm{d}t|$ for each ROI using all the $F_{ana}$ frames $(t)$ and the particles $(i)$ in it at each frame. Then, we plot only the histograms for the ROIs which have the top eight and nine peak probability densities for CFL-induced and alignment-induced bands, respectively. The results are plotted in *Figure 3—figure supplement 3c and d*, respectively. One can find the clear difference in these histograms between the CFL-induced and alignment-induced bands. The peak position and height for the CFL-induced band have large varieties, whereas those for alignment-induced band are less distributed. Furthermore, the peaks for the CFL-induced band are much higher than those for alignment-induced band. *Figure 3i* of the main text and *Figure 3—figure supplement 3e* plot three typical histograms from *Figure 3—figure supplement 3c and d*, respectively.

## Acknowledgements

We are grateful to M Tarama, T Yamamoto and D Sipp for critical reading of this manuscript, and all member of Laboratory for Physical Biology for discussion. This work was supported by JSPS KAKENHI Grant Numbers JP17J05667 (to MH); JP16K17777 and JP19K03764 (to TH); and JP26610129 (to HK).

## Additional information

### Funding

| Funder | Grant reference number | Author |
| --- | --- | --- |
| Japan Society for the Promotion of Science | JP17J05667 | Masayuki Hayakawa |
| Japan Society for the Promotion of Science | JP16K17777 | Tetsuya Hiraiwa |
| Japan Society for the Promotion of Science | JP19K03764 | Tetsuya Hiraiwa |
| Japan Society for the Promotion of Science | JP26610129 | Hidekazu Kuwayama |
| RIKEN | | Tatsuo Shibata |

The funders had no role in study design, data collection and interpretation, or the decision to submit the work for publication.

### Author contributions

Masayuki Hayakawa, Conceptualization, Data curation, Formal analysis, Validation, Investigation, Methodology, Writing - original draft, Writing - review and editing; Tetsuya Hiraiwa, Formal analysis, Funding acquisition, Investigation, Methodology, Writing - original draft, Writing - review and editing, Designed and performed the simulation; Yuko Wada, Investigation; Hidekazu Kuwayama, Resources, Supervision, Funding acquisition, Writing - original draft, Writing - review and editing; Tatsuo Shibata, Conceptualization, Data curation, Formal analysis, Supervision, Funding acquisition, Validation, Investigation, Methodology, Writing - original draft, Project administration, Writing - review and editing

### Author ORCIDs

Masayuki Hayakawa (iD) https://orcid.org/0000-0002-9245-9593
Tetsuya Hiraiwa (iD) https://orcid.org/0000-0003-3221-345X

Hidekazu Kuwayama (iD) https://orcid.org/0000-0002-4362-0790
Tatsuo Shibata (iD) https://orcid.org/0000-0002-9294-9998

## Decision letter and Author response
Decision letter https://doi.org/10.7554/eLife.53609.sa1
Author response https://doi.org/10.7554/eLife.53609.sa2

## Additional files

### Supplementary files
• Transparent reporting form

### Data availability
All data generated or analysed during this study are included in the manuscript and supporting files. Source data files have been provided for Figure 1, 2, 3 and 4.

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
