## [Decision Letter]

**Acceptance summary:**

This manuscript describes a new form of non-chemotactic behavior in the form of multicellular wave propagation in a *Dictyostelium* community. Multiscale analyses including cell tracking and theory describe a mechanism whereby the waves consist of cellular spatial densities that form as single cells penetrate and leave them by a process termed Contact Following Locomotion. While the biological function of this process is yet unclear, the observations reveal how complex and dynamic cellular patterns can form following simple cell-cell interactions.

**Decision letter after peer review:**

Thank you for submitting your article "Polar pattern formation induced by contact following locomotion in a multicellular system" for consideration by *eLife*. Your article has been reviewed by two peer reviewers, including Tâm Mignot as the Reviewing Editor and Reviewer #1, and the evaluation has been overseen by Aleksandra Walczak as the Senior Editor. The following individual involved in review of your submission has agreed to reveal their identity: Sander Tans (Reviewer #2).

The reviewers have discussed the reviews with one another and the Reviewing Editor has drafted this decision to help you prepare a revised submission.

This paper by Shibata et al. explores an intriguing phenomenon of periodic wave formation in *Dictyostelium discoideum*. Using a strain that cannot perform chemotaxis the authors discover a yet undescribed phenomenon where the cells spatially assemble at the mesoscale into high density waves of varying width and persistent directed motion. As it grows over time, this phenomenon propagates until the bands become arranged periodically at the scale of the entire cell community. The authors explore the cellular basis and genetic basis of this phenomenon and come up with the compelling conclusion that the bands form as cell penetrate the band front and escape at the back. Contrarily to the lower density areas, the bands are marked by a high level of polar order resulting from aligned opposite cell flows that move transversal to the wave movement direction. Genetic basis and computational modeling convincingly reveal that a known phenomenon called contact following locomotion (CFL), in which a cell tends to follow another cell upon direct contact, explains the formation and the propagation of the waves. The resulting picture is quite elegant and suggests that a mechanism for wave formation: cells become trapped in specific motile conformations as they enter high polar order regions and escape as the polar order collapses.

The emergent higher-order organization and locomotion that the authors describe has a number of complex and intriguing features, such as the traversal movements of individual cells, that translate in a non-intuitive way to the population level. While complex, the system is still rather clean in the sense that they do not display chemotactic communication. Hence, they are suited for the quantitative active-matter approach chosen by the authors. The paper continues from earlier work in 2013 where the banding phenomenon was also studied, and now provide a highly quantitative analysis. As such, the work has direct value within the active matter community, while the biological relevance of the effects is less direct, though not absent.

Essential revisions:

1) Although the authors get close to nailing an intuitive explanation, they often fall short, which was a bit frustrating. It is not presently clear how single cell movements correlate with the migration direction observed as internal structures in the polar ordered region? The individual cell trajectories, difference in speeds, etc, ultimately 'implies' a critical cell turnover at the band (which is supported by the model), but this is not experimentally demonstrated nor intuitively visualized how this would be central. For example, in the subsection “Analysis of single cell trajectories”, it is mentioned that the cells direction is distributed around the direction of band propagation, but the stripes within the bands extend perpendicular to the direction of propagation. Do the cells that enter these structures follow these flows for sustained period or do they escape quickly mostly traversing the bands? Subsection “Internal structure in the polar ordered region”: the cells are proposed to move inward in one stripe and then downward in the other. This appears contradictory with the single cell trajectory described. Please clarify this picture providing quantifications that cement it would provide a nice example of how something very non-intuitive then becomes transparent.

2) The authors introduce the term Contact Following Locomotion, as if it is a major thing in the Abstract even, but CFL appears to be the same as the previously used term tail-following behavior/locomotion (e.g. subsection “Contact following locomotion is the cell-cell interaction that induces polar pattern formation”, first paragraph, where both are used almost interchangeably). Terminology should be conservative in the community to avoid confusions. Since the two are essentially the same, the same term should be used.

3) It is reasonable to suggest that the *tgrB1* mutant fails to form waves because its ability to perform CFL is reduced. However due to possible pleiotropy of the *tgrB1* mutation other indirect effects could explain why this mutant does not form the pattern. It is agreed that this result with *tgrB1* is informative but it is not a definite genetic proof that CFL is responsible for band formation.

4) The authors noted another interesting observation, namely the transversal movement, which they do not aim to explain within the scope of this paper. At the very least, the authors should provide the best explanation they can.

5) Grammar, Abstract: paradigms do not induce. Also that phrase initially seemed logically wrong, with inhibition of locomotion leading to cells migrating away.

6) Introduction, first paragraph, there are other examples of eukaryotic systems in which contacts lead to higher order organized movement, while here the opposite is suggested. There was a study on migrating cells in circle shaped corrals, and also of rotations of various organoids.

7) Subsection “Internal structure in the polar ordered region”, penultimate sentence: please rephrase.

8) Do the simulations reproduce what is observed in Figure 4A?

9) Discussion, first sentence: novelty is suggested in that phrase but these findings were not demonstrated for the first time here.

10) Discussion, second paragraph: Is the notion that local cell density is not relevant supported by the model?

11) Discussion, third paragraph: 'not seen in the model without CFL'. Please refer to appropriate figure that backs this up. As a more general comment: please go through the paper, identify conclusions, and check if an additional reference to a figure would be helpful. There are other such occasions.

---

## [Author Response]

Essential revisions:1) Although the authors get close to nailing an intuitive explanation, they often fall short, which was a bit frustrating. It is not presently clear how single cell movements correlate with the migration direction observed as internal structures in the polar ordered region? The individual cell trajectories, difference in speeds, etc, ultimately 'implies' a critical cell turnover at the band (which is supported by the model), but this is not experimentally demonstrated nor intuitively visualized how this would be central.

To show the cell turnover more directly, we plot the cell trajectories in the *x-t* coordinate as shown in Figure 2F in the revised manuscript. The figure clearly indicates that cells enter the band at its front and leave the band at its back. This visualization suggests that what is propagating is the cell density profile rather than a group of particular cells, which is an essential property of this traveling band phenomenon.

For example, in the subsection “Analysis of single cell trajectories”, it is mentioned that the cells direction is distributed around the direction of band propagation, but the stripes within the bands extend perpendicular to the direction of propagation. Do the cells that enter these structures follow these flows for sustained period or do they escape quickly mostly traversing the bands? Subsection “Internal structure in the polar ordered region”: the cells are proposed to move inward in one stripe and then downward in the other. This appears contradictory with the single cell trajectory described. Please clarify this picture providing quantifications that cement it would provide a nice example of how something very non-intuitive then becomes transparent.

To clearly show how the analysis of single cell trajectories and the optical flow analysis are compatible, we first reevaluated the trajectories of individual cells used for the analysis shown in Figure 2, and found that the mean migration direction is 176.2 degrees and the standard deviation was 42.7 degrees (Figure 2C). From the optical flow analysis, we also plotted the distribution of migration direction in Figure 3F (black), where the mean and standard deviation of migration direction are 178 and 31 degrees, respectively. Thus, the standard deviations in the migration direction were comparable in these different analyses. These results consistently indicate that the cell migration direction within the traveling band shows a relatively large fluctuation. We also compared the histograms obtained from these two analyses, as shown in Author response image 1, indicating that the two histograms are quite similar to each other. We show the standard deviations in the migration direction in the two analyses in the revised manuscript.

We next examine whether the internal structure found in the optical flow analysis shown in Figure 3C, D is also seen in the trajectories of individual cells. In Figure 3—figure supplement 4, which was newly added to the revised manuscript, the migration direction obtained from cell trajectories is indicated by color as a function of position *x* and time *t*. Cell trajectories are also shown by solid lines. The color-coded migration direction indicates the presence of heterogeneity with light green and dark blue regions, as in the case of Figure 3C, D. Each trajectory does not pass through the same color regions, but move through regions with different colors, implying that cells change their migration directions following the flow directions in the regions. The time interval that cells stay in a stripe structure can be quantitatively evaluated in Figure 3—figure supplement 4 in the revised manuscript. The cell migration speed in the *x*-direction was 0.28±0.09μm/s. Since the width of stripes was estimated to be around 50 to 100 μm (from the optical flow analysis the stripe width was estimate at 125 m as shown in Figure 3—figure supplement 1A), the time scale that cells pass across a stripe is about 200-400 s., if the stripe is almost immobile as we have shown in the main text. The time interval that cells travel across the traveling band is about 1120 s, in the case of Figure 3—figure supplement 1A. We therefore concluded that, while cells pass several different stripes during the time interval that cells stay in the traveling band, the cells stay in each stripe structure for sustained period of time. The analysis shown in Figure 3—figure supplement 4 and the optical flow analysis consistently indicate that there is an internal structure with respect to the migration direction within the traveling band region. We show this additional analysis in the revised manuscript.

2) The authors introduce the term Contact Following Locomotion, as if it is a major thing in the Abstract even, but CFL appears to be the same as the previously used term tail-following behavior/locomotion (e.g. subsection “Contact following locomotion is the cell-cell interaction that induces polar pattern formation”, first paragraph, where both are used almost interchangeably). Terminology should be conservative in the community to avoid confusions. Since the two are essentially the same, the same term should be used.

In the study of collective cell migration of *Dictyostelium* cells, “contact following” has been used to describe the property of the cells to follow the other cells in contact (Shaffer, 1962, Dormann et al., 2002, Umeda and Inouye, 2002, Fujimori et al., 2019). More recently, for the tail-following behavior found in a cultured mammalian epithelial cell, “contact following of locomotion (CFL)” have been proposed in (Li and Wang, 2018), since the behavior is similar to contact following of *Dictyostelium* cell. We decided to use “contact following locomotion (CFL)” following this recent work (Li and Wang, 2018) with minor modification for the grammatical issue. Because of these previous works, it would be appropriate to use “contact following locomotion (CFL)” in this paper. In the revised manuscript, we mention “contact following” in the study of *Dictyostelium* cells with references in Discussion.

3) It is reasonable to suggest that the tgrB1 mutant fails to form waves because its ability to perform CFL is reduced. However due to possible pleiotropy of the tgrB1 mutation other indirect effects could explain why this mutant does not form the pattern. It is agreed that this result with tgrB1 is informative but it is not a definite genetic proof that CFL is responsible for band formation.

We agree that several factors can affect the traveling band formation, such as cell-cell interactions, and locomotive activities of individual cells. As we have shown in the manuscript, the characteristic cell-cell interaction is the CFL, which is reduced in *tgrB1* mutant. We also compare the locomotive activity between control and the mutant, and the velocity auto-correlation function exhibited a similar behavior. These analyses imply that the difference between KI cell and *tgrB1* mutant cell is the ability of CFL. Therefore, CFL is a reasonable candidate of the cell-cell interaction that can explain the traveling band formation. Our mathematical model also supports this idea. Still, we cannot exclude other possibility that could explain the defect in the traveling band formation. We mention this point in the revised manuscript.

4) The authors noted another interesting observation, namely the transversal movement, which they do not aim to explain within the scope of this paper. At the very least, the authors should provide the best explanation they can.

Although the complete analysis remains to be a future problem, we try to provide an explanation on the mechanism of transversal movement. We added the following sentences in the Discussion section:

“A population of cells enters the band at its front with directional alignment induced by CFL in random direction. During the relative movement of these cells from the front to the end of band, the migration direction may not be dampened completely to the direction of band propagation probably due to the directional persistence due to CFL. In this way, subpopulations with respect to the migration direction are formed when CFL is present. A full analysis of this mechanism remains to be a future topic.”

5) Grammar, Abstract: paradigms do not induce. Also that phrase initially seemed logically wrong, with inhibition of locomotion leading to cells migrating away.

We improved the sentence to avoid the grammatical problem. Whereas it is intuitively unobvious if the contact inhibition of locomotion (CIL) generates collective cell migration, CIL has been seen in the collective cell migration of neural crest cells (Carmona-Fontaine et al., 2008). It has been also shown theoretically that a correlated migration of cells can be induced by CIL (Hiraiwa, 2019). Thus, the phrase is not logically wrong.

6) Introduction, first paragraph, there are other examples of eukaryotic systems in which contacts lead to higher order organized movement, while here the opposite is suggested. There was a study on migrating cells in circle shaped corrals, and also of rotations of various organoids.

According to the reviewers’ suggestion, we mention the studies on the organized motion on circular micropatterns and spheroids in the Introduction of the revised manuscript. However, as far as we know, there is no report that establishes the connection between traveling band formation of eukaryotic cells and the microscopic cell-cell interactions that induce the traveling band formation. We mention this point more clearly in the first paragraph of Introduction.

7) Subsection “Internal structure in the polar ordered region”, penultimate sentence: please rephrase.

We have rephrased the sentence in this line. In particular, we explain the reason of why the cell moves from the front to the end of band in a reference frame co-moving with the band. The reason is that the speed of traveling band is faster than the speed of cells. Then, the transversal motion is explained during the relative motion of cells within the traveling band.

8) Do the simulations reproduce what is observed in Figure 4A?

For the same parameter values shown in Figure 3G, H, we performed a simulation with low cell density, as shown in Author response image 2. Contact following locomotion (CFL) can be seen between two to four cells. However, we found that the life-time of CFL in the simulation was shorter than that in the experiments. This is an aspect that the model cannot reproduce and is considered a future issue.

**Author response image 2. respfig2:** 

9) Discussion, first sentence: novelty is suggested in that phrase but these findings were not demonstrated for the first time here.

We now refer the previous work by one of the authors in this sentence.

10) Discussion, second paragraph: Is the notion that local cell density is not relevant supported by the model?

We found that the width of band and the order parameter within the band exhibited a positive correlation (Figure 3B). Because the order parameter is expected to increase with the cell density in general, one possibility for the correlation between the width and the order parameter could be that the cell density increases with the band width. However, our data indicates the local cell density within the band is similar across different samples with different width (Figure 1—figure supplement 1C), indicating that the above hypothesis does not hold.

In the model, we changed the values of several parameters, such as the effect of CFL ζ, alignment effect α, and the cell density ρ of entire area. For these simulations, the cell density within the traveling band was almost constant (Author response image 3), supporting our experimental observation.

**Author response image 3. respfig3:** 

In the simulations, the width of band depended on some parameters such as the strength of alignment effect α (Figure 3—figure supplement 2).Nevertheless, we should also note that the system size in our simulation may not be sufficiently large to see the change in the width of band systematically. It has been reported that, for the well-studied simple model of a motile element system, the polar order region is split into multiple sequential bands without much change in the band width as long as the system size is large enough (Solon et al., 2015). Thus, the system size may strongly affect the band width. Due to this complexity, it is difficult to strongly conclude that in our simulation the cell density is constant for bands with different width.

In addition to this, so far, we have not found a reason of why the band width shows such variation in the experiment and how it is correlated with the order parameter. In this respect, it is difficult for the simulation to reproduce the correlation between band width and the order parameter, and thus to test if the density is constant for different band width in the simulation.

11) Discussion, third paragraph: 'not seen in the model without CFL'. Please refer to appropriate figure that backs this up. As a more general comment: please go through the paper, identify conclusions, and check if an additional reference to a figure would be helpful. There are other such occasions.

We have added references to figures to the sentences throughout the paper.